# Carbon Release Characteristics at Soil–Air Interface under Litter Cover with Different Decomposition Degrees in the Arbor and Bamboo Forests of Pi River Basin

Junwei Zhang [1,2], Tao Du [3,4], Shanshan Liu [2], Sintayehu A. Abebe [5], Sheng Yan [2,6], Wei Li [2] and Tianling Qin [2,*]

1 College of Hydraulic & Environmental Engineering, China Three Gorges University, Yichang 443002, China; zjw920126@163.com
2 State Key Laboratory of Simulation and Regulation of Water Cycle in River Basin, China Institute of Water Resources and Hydropower Research, Beijing 100038, China; liushanshan198705@163.com (S.L.); yans19900814@163.com (S.Y.); weil01234@163.com (W.L.)
3 School of Water Conservancy and Hydroelectric Power, Hebei University of Engineering, Handan 056038, China; dutao@mwr.gov.cn
4 Chinese Hydraulic Engineering Society, Beijing 100053, China
5 Hydraulic and Water Resources Engineering Department, Debre Markos University Institute of Technology, Debre Markos 269, Ethiopia; sentaddis@gmail.com
6 College of Conservancy Engineering, Zhengzhou University, Zhengzhou 450000, China
* Correspondence: qintl@iwhr.com

**Abstract:** This study adopted the method of "exchanging space for time" and set up three experimental groups based on the shape, degree of damage, and degree of humification of the litter, namely the undecomposed layer, the semi-decomposed layer, and the decomposed layer. Using typical slopes of arbor and bamboo forests in the Pi River Basin as the research object, from October 2021 to December 2022, the soil carbon release flux was measured by using a closed static chamber gas chromatography method to reveal the carbon release law at the soil–air interface during the decomposition process of litter and quantitatively characterize the dynamic impact of the litter decomposition process on soil carbon release flux. Results showed that soil methane flux remained negative (sink) while soil carbon dioxide flux was positive (source) in both litter-covered and bare soil conditions. The methane and carbon dioxide release from soil was positively correlated with and significantly influenced by environmental factors such as soil moisture content and temperature. The methane release flux from soil showed a linear fitting relationship with soil moisture content and temperature, while the carbon dioxide release flux from soil was more in line with the exponential fitting relationship with soil moisture content and temperature. However, there were significant differences in the roles of various factors under different types of litter.

**Keywords:** Pi River basin; soil–air interface; carbon flux; soil moisture content

## 1. Introduction

Litter is an essential component of terrestrial ecosystems, playing an important role in nutrient and carbon cycling between soil and atmosphere. It is a product of metabolism during the growth process of vegetation and an important channel connecting vegetation and soil. The decomposition process is a material cycle dominated by carbon elements. As the decomposition progresses, the net carbon content in the litter decreases overall, but there are certain fluctuations at different periods [1]. The litter–soil interface is the most active and complex part of underground ecosystems and the core site of carbon cycling. It is generally believed that effective nutrients gradually change from the surface layer of litter to the lower layer of soil [2]. The litter covering the soil surface releases a large amount of carbon during the decomposition process, which has a certain impact on the release, migration, and transformation of soil carbon (Figure 1). Therefore, quantitatively identifying the

impact of carbon produced by litter decomposition on soil carbon release, identifying the correlation between soil water and soil carbon during litter decomposition, and deriving the simulation of soil–carbon–water process changes under litter cover conditions are key scientific issues in analyzing the impact of litter on soil carbon flux.

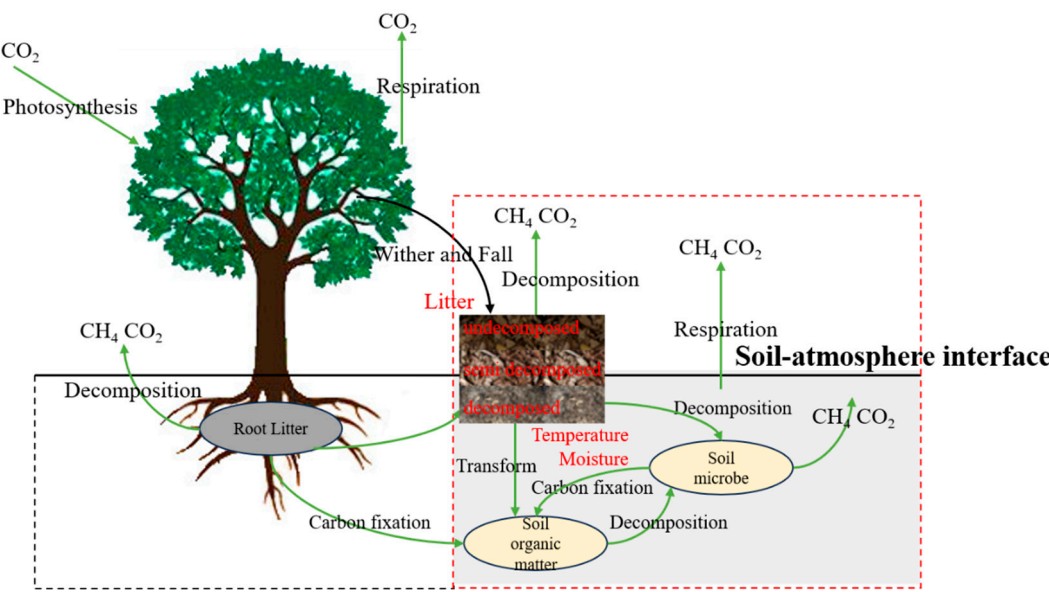

**Figure 1.** Effects of litter decomposition on carbon flux process at litter–soil interface.

Litter is the surface soil's main source of organic matter [3]. The aboveground biomass promotes more litter input and positively impacts the standing stock of litter. Due to the stimulation of more aboveground biomass input, soil organic matter also positively impacts the standing stock of litter [4]. The results show that all nutrients in litter are significantly positively correlated with soil. In contrast, soil bulk density and particle density are significantly negatively correlated with litter carbon content and positively correlated with total porosity by using the decomposition bag method to understand nutrient differences after litter decomposition [5]. Onwuka and Mang have emphasized the importance of soil temperature in litter decomposition rate, organic matter mineralization, soil water content, plant hydraulic conduction, and evapotranspiration, and soil temperature is one of the key abiotic variables affecting soil processes [6]. Soil respiration is significantly influenced by soil temperature, soil moisture content, litter removal, and root system. It has strong seasonal variation and decreases with increasing soil depth, which is exponentially correlated with soil temperature. As the soil moisture content increases, it increases until the threshold is reached and then decreases. The effects of aboveground litter and plant roots on soil respiration are also different [7]. Studies have shown that, in temperate forests, litter decomposition accounts for approximately 5% to 45% of the total soil $CO_2$ emissions [8,9]. Rubino et al. have used $^{13}C$ isotope tracer technology to reveal the continuous mineralization and transformation mechanism of carbon in litter, and found that two thirds of the carbon in litter entered the soil as soluble carbon or debris, and one third entered the atmosphere as $CO_2$ [10]. Lu et al. have studied the soil $CO_2$ release rate of the Minjiang River estuary wetland by adding different types of litter. The results have shown that litter addition treatment significantly promoted the soil $CO_2$ release rate of the wetland and significantly increased the contents of dissolved organic carbon and microbial biomass carbon in the Minjiang River estuary wetland [11]. Zhao et al. have studied soil respiration by changing the input mode of litter and found that the removal of litter significantly reduced soil microbial biomass, thus reducing microbial respiration (13%) and soil respiration (14%), but the increase of litter has no effect [12].

Research on forest litter layer and soil layer has mostly been conducted separately in the past, and a relatively small number of studies consider the two as a continuum of matter

and energy [13,14], especially the changes of carbon water relationship at the interface under the influence of litter. According to the division of vegetation zoning in China, the Pi River basin belongs to a mixed forest zone of deciduous and evergreen coniferous and broad-leaved arbors, with a vegetation coverage rate of 77.3% [15]. The vegetation types are mainly arbor and bamboo forests. In this study, the characteristics of carbon release at the litter–soil interface and the rule of carbon release at the soil–air interface during litter decomposition were revealed by taking typical slopes of arbor and bamboo forests in the Pi River basin as research objects. The dynamic effects of the litter decomposition process on soil carbon release flux were quantified and improved the understanding of carbon dynamic changes at the litter–soil interface and the carbon cycling mechanism in terrestrial ecosystems. It also provided a theoretical basis for accurately predicting the response of the terrestrial carbon cycle to future climate change.

## 2. Materials and Methods

### 2.1. Study Area

This study was conducted at the Pi River basin (Figure 2) in Anhui province, China, located at 30°55′~31°23′ N and 115°46′~116°30′ E. The Pi River basin belongs to the Huaihe River and covers an area of 1816.83 km². The elevation in the basin varies between 84 and 1740 m. The comparison of soil vegetation properties between arbor and bamboo forests is shown in Table 1. The Pi River basin belongs to the humid monsoon climate zone of the North Subtropical Zone. It is influenced by the monsoon, the Western Pacific subtropical high pressure, and the continental cold high pressure, forming four distinct climate characteristics of "warm spring, hot summer, cool autumn, and cold winter". The annual average temperature of the entire basin is 15.1 °C. The lowest average temperature in January is 3 °C, and the highest in July is 27.9 °C. There are fewer periods of severe cold and heat, and the temperature is suitable. The average annual rainfall in the basin is 1505.4 mm, with uneven rainfall distribution within the year. The average annual rainfall during the flood season is 995.6 mm, accounting for 66.1% of the average annual rainfall.

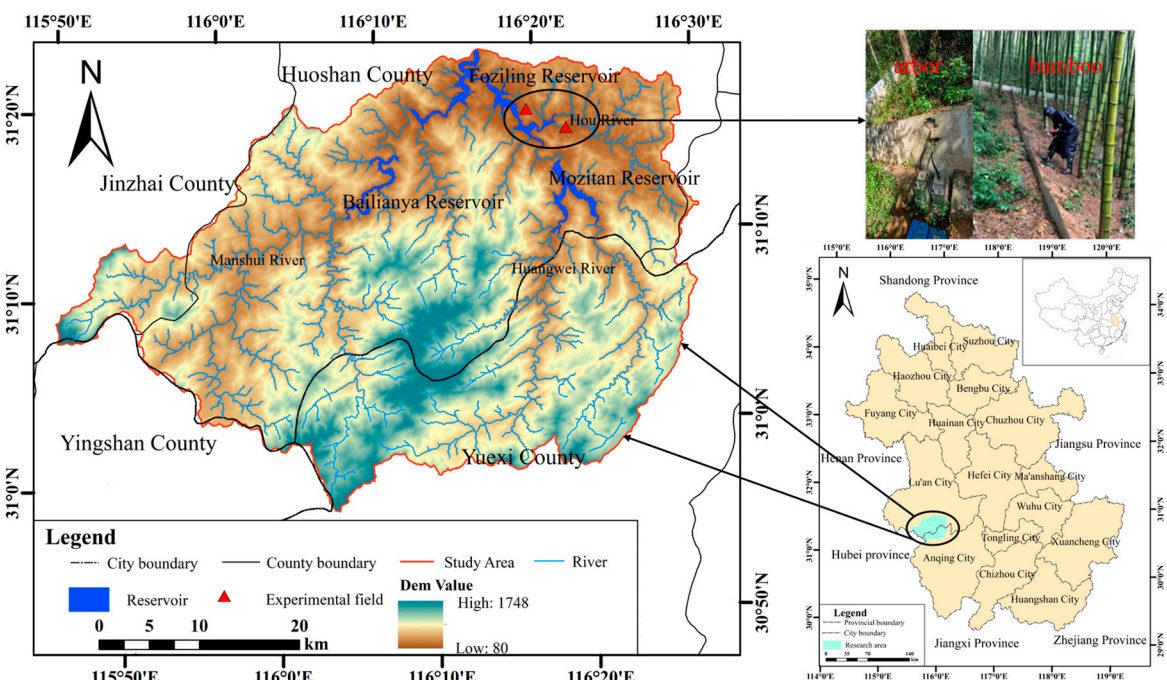

**Figure 2.** Location of the study area.

**Table 1.** Comparison of soil and vegetation properties between arbor and bamboo forests.

| Project | Vegetational Form | Soil Texture | Volume of Litter Storage (t/hm²) | | pH | Appearance Density | Organic Carbon |
|---------|-------------------|--------------|-----------------------------------|--|----|--------------------|----------------|
| | | | Undecomposed Layer | Decomposition Layer | / | g/cm³ | g/kg |
| Arbor | cedar, robur | sandy clay loam, sandy loam | 2.88 | 17.9 | 5.30 | 1.40 | 16.03 |
| Bamboo | Moso bamboo | sandy loam | 3.36 | 8.15 | 5.54 | 1.58 | 19.30 |

### 2.2. Sample Collection and Analysis Methods

This study used a closed static chamber gas chromatography method to measure soil carbon flux from October 2021 to December 2022. A chamber with known volume, bottom area, and stable chemical properties was inserted into the ground, and an appropriate amount of gas was extracted regularly. The concentration of the collected gas was analyzed and measured using a gas chromatograph or gas analyzer. Finally, the flux of the gas on the covered ground was calculated based on the rate of change in concentration over time (Figure 3).

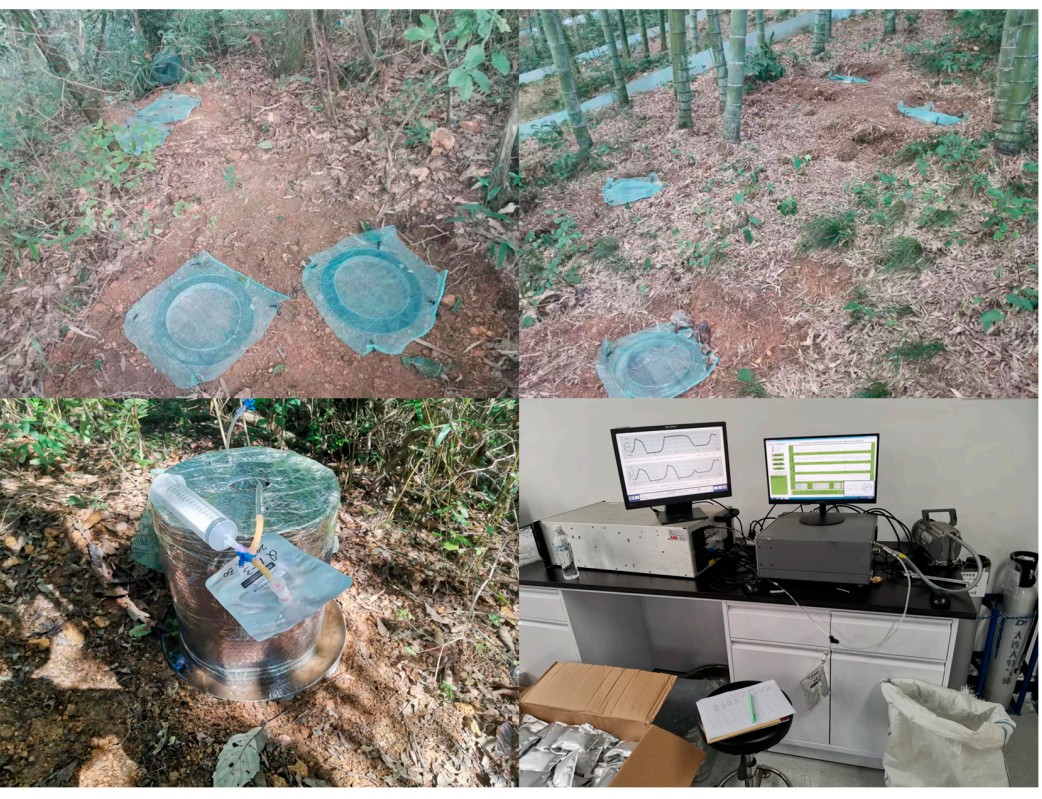

**Figure 3.** Field sampling and laboratory analysis.

Experimental design: Three experimental groups are created, based on the shape, degree of damage, and degree of humification of the litter as follows: undecomposed layer, semi-decomposed layer, and decomposed layer, as well as a control group without litter. After drying to a constant weight, the litter is added to the surface of the bare soil. After a period of time, the experiment begins when the surrounding environment stabilizes.

Specific sampling steps: Before sampling, place the chamber mouth upwards for about 5 min to allow the chamber to be filled with air. During sampling, place the sampling chamber on the ground and insert the chamber mouth into the ground to ensure that the air inside is isolated from the outside. When the flux chamber is first inserted into the

ground, 100 mL of ground air (as the background concentration) is collected. Then, 100 mL of gas in the chamber is extracted five times every 8 min (3 parallel samples are taken each time). The frequency of measurements is once every three days, from 9:00 to 11:00 in the morning. The gas sample is injected into an air bag with a volume of 0.3 L (produced by Delin Air Bag Factory in Dalian, China). After the gas samples are brought back to the laboratory, the concentrations of $CH_4$ and $CO_2$ are analyzed and detected by using a greenhouse gas analyzer (PICARRO, G2301, Santa Clara, CA, USA; detection range: $CH_4$ 0–20 ppm, $CO_2$ 0–1000 ppm, detection interval < 5 s, response time < 3 s). Based on the detection results, the rate of gas concentration change over time at each monitoring point is plotted. Among them, the measured value of clearing surface litter treatment is the community soil respiration rate ($R_S$). The measured value of not clearing the surface litter treatment is the sum of soil respiration rate and the $CO_2$ release rate of litter decomposition ($R_{S+L}$). The difference is the $CO_2$ release rate of litter decomposition ($R_L$).

*2.3. Calculation Method of $CO_2$ and $CH_4$ Diffusion Flux*

Gas release flux refers to the change in a gas's concentration per unit area over time. A positive value indicates a gas emission from the soil into the atmosphere. In contrast, a negative value indicates the gas absorption the soil into the atmosphere. This study used the following formula to calculate the release flux [16]:

$$Flux = \frac{slope \times F_1 \times F_2 \times volume}{surface \times F_3}$$

where *Flux* was the gas flux at the soil–air interface[mg·($m^2$·h)$^{-1}$]; *slope* was the rate of change of gas concentration in the chamber with time; $F_1$ was the conversion coefficients from μL·L$^{-1}$ to μg·m$^{-3}$ (1798.45 μg·m$^{-3}$ for $CO_2$ and 655.47 μg·m$^{-3}$ for $CH_4$); $F_2$ was the conversion coefficient from minute to hour (60); *volume* was the volume of gas in the static tank ($m^3$); *surface* was the area of the bottom of the flux chamber ($m^2$); $F_3$ was the conversion coefficients from μg to mg (1000).

*2.4. Physical and Chemical Factors*

The FDR (frequency domain reflection) soil volume moisture probe is buried in the experimental area to monitor soil moisture content and temperature. One meteorological station is set up outside the forest to monitor environmental factors such as rainfall, temperature, air pressure, humidity, and wind speed. The observation is automatically recorded by the meteorological station system, with a recording frequency of 1 h per time. The pH of the soil is measured using an acidity meter. Soil and litter samples are mainly analyzed for organic carbon content using an organic carbon analyzer (Jena, Germany, 3100).

*2.5. Data Statistical Analysis*

The data types obtained during the experiment mainly include soil carbon release, litter water holding capacity, litter moisture content, surface soil moisture content, soil volume moisture content at different depths, soil temperature, air temperature, humidity, and other related data. Excel 2016 is used to organize and analyze the initial data, and SPSS 21 software is used to perform correlation analysis and fitting on the relevant data. The charts are mainly drawn by using Origin2018 and Grapher10 software.

## 3. Results

*3.1. Variation Characteristics of Physical and Chemical Factors*

As shown in Table 2, under the condition of no litter cover, the soil moisture content of the arbor is between 8.18% and 17.48%, while the average soil moisture content of bamboo forests is 24.62%. Under litter coverage, the range of soil moisture content variation in arbor is from 7.31% to 22.27%, and the average soil moisture content in bamboo forests is 26.61%. The overall performance of the soil moisture content under the cover of litter is generally

higher than without litter. The trend of soil temperature change is similar to that of soil moisture content change.

**Table 2.** Variation of soil temperature and soil water under different decomposition levels of litter cover.

| Project | | Arbor | | | | Bamboo | | | |
|---|---|---|---|---|---|---|---|---|---|
| | | No Litter | Undecomposed Layer Litter | Semi-Decomposed Layer Litter | Decomposition Layer Litter | No Litter | Undecomposed Layer Litter | Semi-Decomposed Layer Litter | Decomposition Layer Litter |
| soil temperature (°C) | average | 16.9 | 17.2 | 17.7 | 18.2 | 15.7 | 16.1 | 16.7 | 17.7 |
| | max | 26.0 | 26.2 | 26.5 | 27.3 | 24.2 | 24.5 | 26.0 | 28.2 |
| | minimum | 3.5 | 3.5 | 3.6 | 3.6 | 4.2 | 4.4 | 4.5 | 4.5 |
| | rangeability | 22.5 | 22.7 | 22.9 | 23.7 | 20.0 | 20.1 | 21.5 | 23.7 |
| Project | | no litter | | litter | | no litter | | litter | |
| soil water (%) | average | 13.35 | | 17.44 | | 24.62 | | 26.61 | |
| | max | 17.48 | | 22.27 | | 29.61 | | 35.20 | |
| | minimum | 8.18 | | 7.31 | | 17.98 | | 17.47 | |
| | rangeability | 9.30 | | 14.96 | | 11.63 | | 17.73 | |

*3.2. Variation Characteristics of CH$_4$ Flux at Soil–Air Interface under Different Decomposition Levels of Litter Cover*

As shown in Figure 4, regardless of whether the soil surface was covered with litter, the methane release flux at the soil–air interface of arbor and bamboo forests showed a negative value, manifested as a sink of atmospheric methane, and the soil consumes atmospheric methane. The main reason for this phenomenon was that the soil was oxygen-rich, and methane-oxidizing bacteria continuously oxidized and consumed atmospheric methane. The average consumption of atmospheric methane in forests without litter cover was 0.250 mg·m$^{-2}$·h$^{-1}$, and that with an undecomposed layer litter cover was 0.174 mg·m$^{-2}$·h$^{-1}$. The average methane consumption of soil covered by litter in the semi-decomposition layer and decomposition layer was 0.208 mg·m$^{-2}$·h$^{-1}$ and 0.221 mg·m$^{-2}$·h$^{-1}$, respectively. The average methane consumption of bamboo forests without litter cover was 0.114 mg·m$^{-2}$·h$^{-1}$, and the atmospheric methane consumption of soil covered with litter of different decomposition degrees was in the order of semi-decomposed layer > undecomposed layer > decomposed layer. The consumption was 0.183 mg·m$^{-2}$·h$^{-1}$, 0.113 mg·m$^{-2}$·h$^{-1}$ and 0.088 mg·m$^{-2}$·h$^{-1}$, respectively.

The results showed that the overall methane consumption at the soil–air interface significantly decreased compared to the conditions without litter after covering different decomposed litters. The average methane consumption in the arbor forest decreased by 19.6%, and the methane consumption under the decomposition layer litters of the bamboo forest decreased by about 22.4%. There was little difference in methane consumption under the condition of undecomposed litter, which was consistent with the research results in temperate and subtropical areas [8,17,18].

As shown in Figure 5, the addition of litter with different degrees of decomposition on the soil surface significantly reduced the methane consumption at the soil–air interface, and the reduction was significantly affected by the degree of litter decomposition, especially in arbor forest (Figure 5A). After adding an undecomposed layer, semi-decomposed layer, and decomposed layer of litter to the soil surface of the arbor forest, methane consumption decreased by 0.088 mg·m$^{-2}$·h$^{-1}$, 0.043 mg·m$^{-2}$·h$^{-1}$, and 0.029 mg·m$^{-2}$·h$^{-1}$, respectively. The addition of litter with different degrees of decomposition in bamboo forests had a significant difference in methane consumption at the soil–air interface (Figure 5B). After adding an undecomposed layer of litter, the soil's methane consumption was relatively stable compared to that without litter, with little fluctuation, indicating that the litter's impact was insignificant. After adding semi-decomposed layer litter, the soil's consumption of atmospheric methane significantly increased, while the soil's methane consumption in the atmosphere decreased by 0.0255 mg·m$^{-2}$·h$^{-1}$ after adding decomposition layer litter.

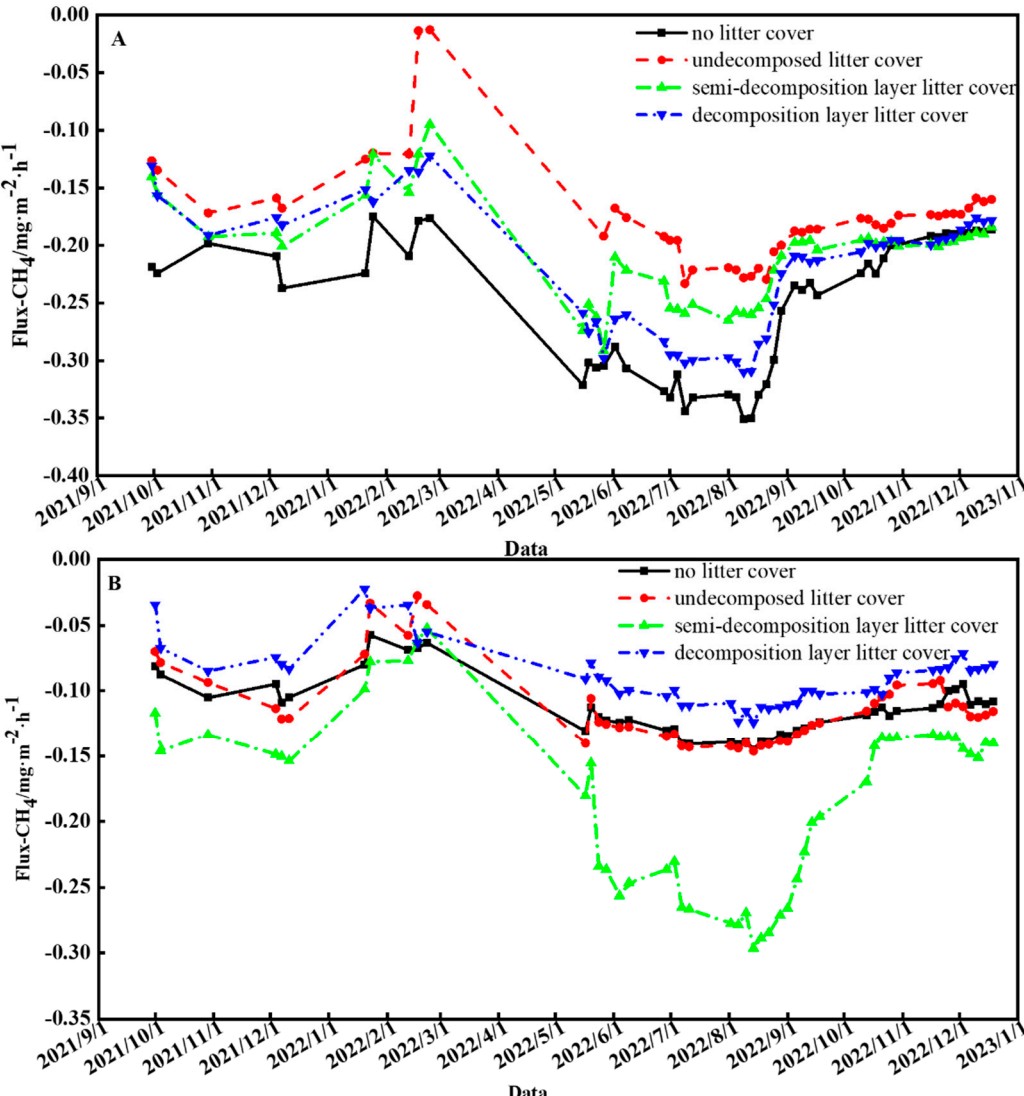

**Figure 4.** The soil–gas interface methane emission flux of arbor (**A**) and bamboo (**B**) under different litter decomposition degrees.

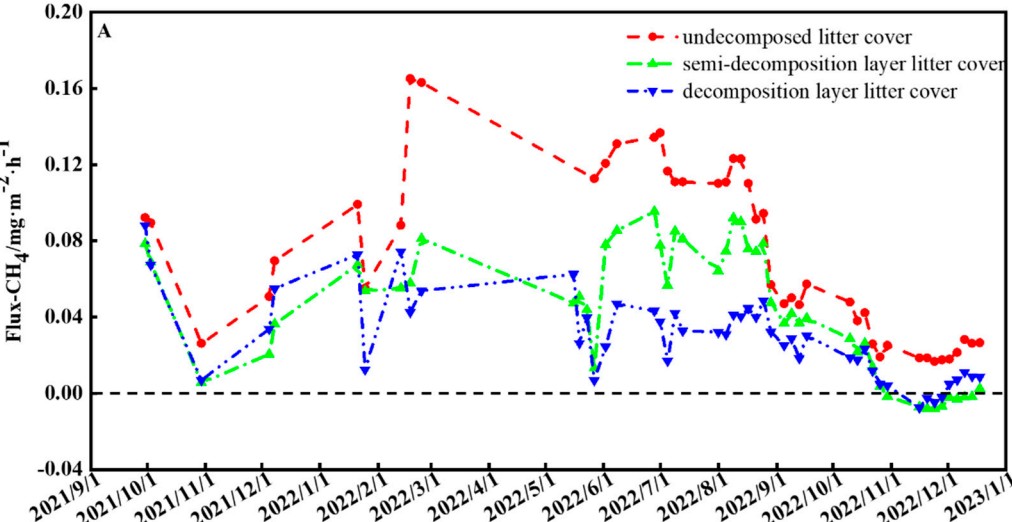

**Figure 5.** *Cont.*

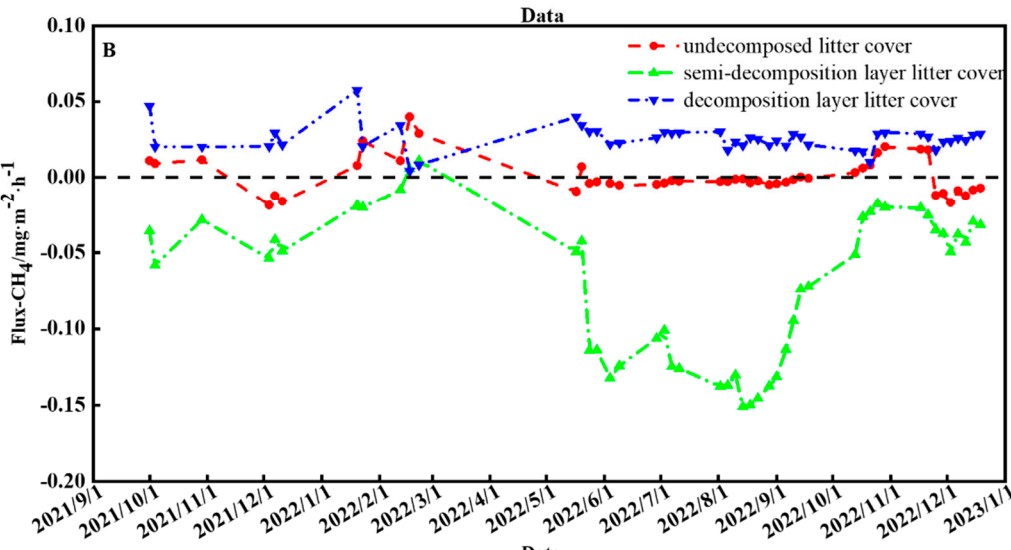

**Figure 5.** The effects of different decomposition degrees of litter on methane release flux at soil–air interface in arbor forest (**A**) and bamboo forest (**B**).

The main reason for this phenomenon was that the litter layer had a natural barrier effect on soil carbon emissions. A layer of impermeable protective film was formed between the litter layer and the soil [19], which blocked or slowed down the gas exchange rate and heat exchange between the atmosphere and the soil, effectively reducing the rate of methane supply from the atmosphere to the soil, and the rate of methane oxidation and consumption in the soil [20]. Especially for undecomposed layer litters, due to their complete shape, specifications, and dimensions, large soil coverage area, and significant barrier effect of litters, when the soil surface was covered with undecomposed layer litters, the consumption of atmospheric methane was significantly lower than that of other decomposition level litters.

*3.3. Variation Characteristics of $CO_2$ Flux at Soil-Air Interface under Different Decomposition Levels of Litter Cover*

As shown in Figure 6, regardless of whether the soil surface was covered with litter, the carbon dioxide release flux at the soil–air interface showed a positive value, representing the source of atmospheric carbon dioxide the release of carbon dioxide from the soil to the atmosphere. Figure 6A shows that after the arbor forest was covered with different degrees of litter decomposition, the carbon dioxide release increased with the intensification of litter decomposition. After adding different degrees of decomposition of litter, the average carbon dioxide release fluxes at the soil–air interface were 480.6 mg·m$^{-2}$·h$^{-1}$, 881.4 mg·m$^{-2}$·h$^{-1}$, and 1025.2 mg·m$^{-2}$·h$^{-1}$, respectively, that was, undecomposed layer < semi decomposed layer < decomposed layer. However, there was a significant difference in the trend of changes in bamboo forests compared to arbor forests (Figure 6B). After adding an undecomposed litter layer, the carbon dioxide release was 809.6 mg·m$^{-2}$·h$^{-1}$. After adding semi-decomposed layer and a decomposed layer litter, the carbon dioxide release was 696.7 mg·m$^{-2}$·h$^{-1}$ and 573.2 mg·m$^{-2}$·h$^{-1}$, respectively. The overall performance was that, as the degree of litter decomposition intensified, the carbon dioxide release gradually decreased.

The addition of litter on the soil surface significantly impacted the release of carbon dioxide at the soil–air interface. As shown in Figure 7A, the soil surface of the arbor forest was covered with semi-decomposed layer and decomposed layer litter. The carbon dioxide released by litter decomposition is 327.9 mg·m$^{-2}$·h$^{-1}$ and 471.7 mg·m$^{-2}$·h$^{-1}$, respectively, accounting for 37.21% and 46.02% of the carbon dioxide release. However, after adding an undecomposed layer of litter, the amount of carbon dioxide released was lower than that released without covering the litter, and there was a significant difference between

the two in the initial stage. This difference showed a gradually smaller trend as the litter continued to decompose. From May to October, the carbon dioxide emissions from bamboo forests (Figure 7B) under litter cover were higher than those without litter cover, while other monitoring times showed that the carbon dioxide emissions under litter cover were lower than those without litter cover.

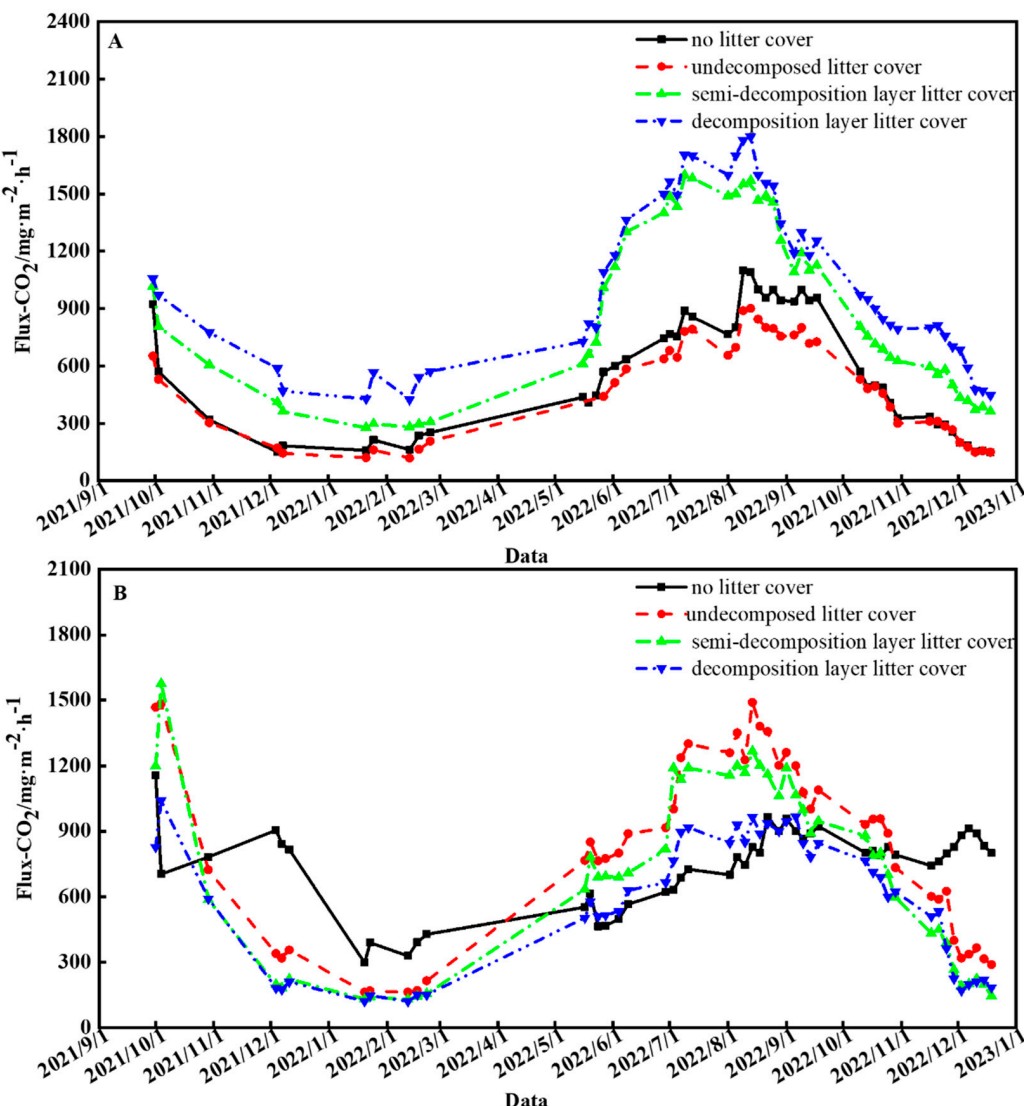

**Figure 6.** The soil–gas interface carbon dioxide emission flux of arbor (**A**) and bamboo (**B**) under different litter decomposition degrees.

This phenomenon did not mean that adding an undecomposed layer of litter on the soil surface absorbed carbon dioxide from the atmosphere. The main reason was that the carbon dioxide generated by the litter decomposition in this study had a relative value, and the litter layer on the soil surface had a certain barrier effect [19]. Compared to the semi-decomposition and decomposition layers, the shape of the undecomposed layer litter was complete, which somewhat hindered the release of carbon dioxide from the soil to the atmosphere. As the litter continues to decompose, the barrier effect of the litter layer gradually decreases. The research results were consistent with those of Yuan Hongye and Yang Jisong et al. [19,21]. The decomposition of litter in forest ecosystems was one of the important sources of surface carbon dioxide emissions. In temperate forests, litter layer decomposition contributed 5% to 45% of total soil carbon dioxide emissions [8,9], while in subtropical forests, litter decomposition released 18% to 42% of total surface soil carbon

dioxide emissions [18,22–25]. In this study, after covering litter with different degrees of decomposition, the carbon dioxide released during the litter decomposition process accounted for 41.61%. Previous studies have shown that soil carbon dioxide emissions are positively correlated with soil temperature, moisture, and soil organic carbon [26,27], and are regulated by soil microbial activity [28].

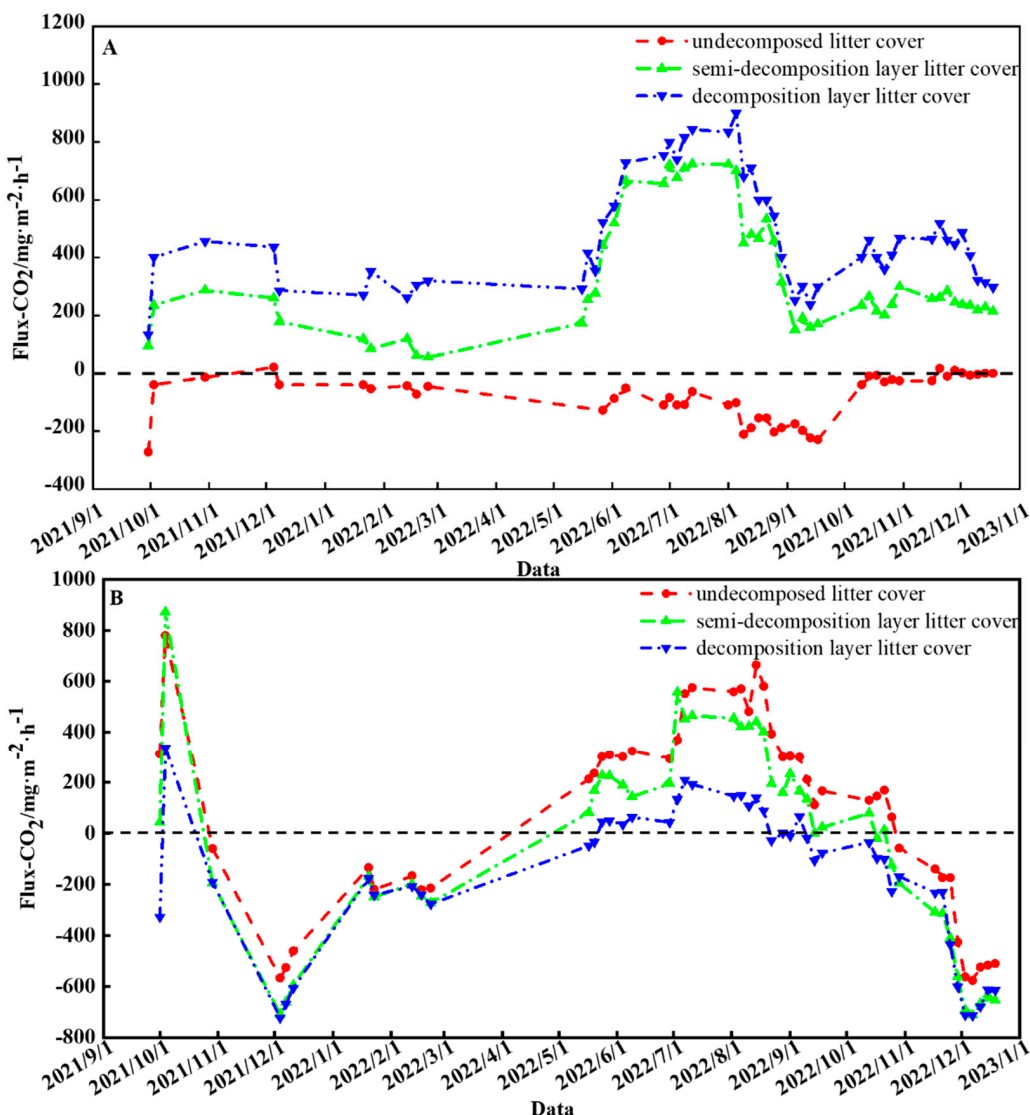

**Figure 7.** Effects of different decomposition degrees of litter cover on carbon dioxide release flux at soil–air interface in arbor forest (**A**) and bamboo forest (**B**).

### 3.4. The Results of Soil Carbon Release Flux under Litter Cover

By averaging data over a year (Table 3), the average methane consumption and carbon dioxide release in the soil under the cover of arbor litter in the Pi River basin were 0.201 mg·m$^{-2}$·h$^{-1}$ and 795.733 mg·m$^{-2}$·h$^{-1}$, respectively. The average methane consumption and carbon dioxide release under the cover of bamboo litter were 0.128 mg·m$^{-2}$·h$^{-1}$ and 693.167 mg·m$^{-2}$·h$^{-1}$, respectively. From a climate perspective (Table 4), the Pi River basin belongs to the subtropical humid monsoon climate zone. Compared with other regions, soil methane consumption was slightly higher than in temperate forest soil. The carbon dioxide release from soil under litter cover was relatively high compared to research results in the same subtropical climate region, and there was no significant difference in soil carbon dioxide release from tropical regions. Overall, methane was a sink of the atmosphere, and carbon dioxide was a source of the atmosphere.

**Table 3.** Carbon release flux at the soil–air interface under litter cover of arbor and bamboo forests.

| Project | Type | No Litter | Under Litter | Undecomposed Layer | Semi Decomposed Layer | Decomposition Layer |
|---|---|---|---|---|---|---|
| $CH_4$ $mg \cdot m^{-2} \cdot h^{-1}$ | arbor | −0.250 | −0.201 | −0.174 | −0.208 | −0.221 |
| | bamboo | −0.114 | 0.128 | −0.113 | −0.183 | −0.088 |
| $CO_2$ $mg \cdot m^{-2} \cdot h^{-1}$ | arbor | 553.400 | 795.733 | 480.600 | 881.400 | 1025.200 |
| | Bamboo | 735.800 | 693.167 | 809.600 | 696.700 | 573.200 |

Note: Under litter is the average of three results: undecomposed layer, semi-decomposed layer, and decomposed layer.

**Table 4.** Soil carbon release fluxes under litter cover in different climatic zones.

| Area | Climatic Zone | Type of Litter | Carbon Release Fluxes | | Reference |
|---|---|---|---|---|---|
| | | | Flux-$CH_4$/ $mg \cdot m^{-2} \cdot h^{-1}$ | Flux-$CO_2$/ $mg \cdot m^{-2} \cdot h^{-1}$ | |
| Brazil | tropic | evergreen broad-leaved forest | - | 728.85 | [29] |
| Hawaii | tropic | evergreen broad-leaved forest | - | 652.25 | [30] |
| Dinghu Mountain | subtropics | evergreen broad-leaved forest | - | 475.83 | [24] |
| Amazon | temperate zone | deciduous forest | −0.01~−0.16 | - | [31] |
| Gongga Mountain | Transition zone between subtropical and temperate zones | fir forest | −0.080 ± 0.066 | - | [32] |
| Hangzhou West Lake Region | subtropics | evergreen broad-leaved forest | - | 275.34 | [33] |
| | | phyllostachys pubescens forest | - | 351.26 | |
| | | tea garden | - | 325.91 | |
| PI River Basin | subtropics | arbor forest | −0.20 | 795.42 | the study |
| | | bamboo forest | −0.13 | 693.20 | |

*3.5. Analysis Results of Soil Carbon Release with Temperature and Soil Moisture Content*

3.5.1. The Analysis Results of Soil Carbon Release with Temperature

By fitting methane and carbon dioxide fluxes to temperature (Table 5), the results showed that temperature had an overall promoting effect on soil methane absorption. In arbor forests, the effect of temperature on soil methane absorption gradually decreased with the deepening of litter decomposition. However, in bamboo forests, except for the semi-decomposition layer, there was no significant difference under other experimental conditions. As shown in Table 6, there was a significant positive correlation between carbon flux and temperature. There was a significant exponential function relationship between carbon dioxide release flux and soil temperature in arbor and bamboo forests (except for the condition without litter cover in bamboo forests), and the temperature sensitivity of soil respiration in arbor forests decreased continuously with the deepening of litter decomposition. In bamboo forests, the temperature sensitivity was also higher in the semi-decomposition layer, except for the conditions without litter cover. The temperature sensitivity under other decomposition degrees of litter cover conditions in bamboo forests was similar to the relevant research results of the Qianjiangyuan Research Station in Zhejiang at the same latitude [34].

**Table 5.** Fitting analysis of methane and carbon dioxide emission fluxes and temperature.

| Project | Experiment Condition | Flux-CH$_4$ | | Flux-CO$_2$ | | |
| | | Fitting Formula | R$^2$ | Fitting Formula | Q$_{10}$ | R$^2$ |
|---|---|---|---|---|---|---|
| Arbor | No litter cover | F = −0.0056T − 0.15 | 0.51 | F = 121.07e$^{0.0798T}$ | 2.22 | 0.81 |
| | Undecomposed litter cover | F = −0.0039T − 0.107 | 0.47 | F = 109.32e$^{0.0766T}$ | 2.15 | 0.84 |
| | Semi-decomposition layer litter cover | F = −0.0034T − 0.148 | 0.41 | F = 235.6e$^{0.0675T}$ | 1.96 | 0.88 |
| | Decomposition layer litter cover | F = −0.0005T − 0.132 | 0.54 | F = 387.78e$^{0.0491T}$ | 1.63 | 0.82 |
| Bamboo | No litter cover | F = −0.0024T − 0.075 | 0.63 | F = 583.97e$^{0.0122T}$ | 1.13 | 0.05 |
| | Undecomposed litter cover | F = −0.0026T − 0.072 | 0.39 | F = 163.53e$^{0.0877T}$ | 2.40 | 0.85 |
| | Semi-decomposition layer litter cover | F = −0.0071T − 0.064 | 0.65 | F = 100.18e$^{0.1007T}$ | 2.73 | 0.87 |
| | Decomposition layer litter cover | F = −0.002T − 0.052 | 0.49 | F = 108.32e$^{0.0828T}$ | 2.29 | 0.86 |

Note: Q$_{10}$ represented temperature sensitivity, which refers to the multiple increase in R for every ten °C increase in temperature; The higher the Q$_{10}$ value, the more sensitive the decomposition of soil organic matter was to temperature changes.

**Table 6.** The correlation between carbon flux and temperature at the soil–air interface.

| Project | | Flux-CO$_2$ | Flux-CH$_4$ | Air Temperature | Soil Temperature |
|---|---|---|---|---|---|
| Arbor | Flux-CO$_2$ | 1 | 0.571 ** | 0.837 ** | 0.785 ** |
| | Flux-CH$_4$ | | 1 | 0.604 ** | 0.615 ** |
| | air temperature | | | 1 | 0.984 ** |
| | soil temperature | | | | 1 |
| Bamboo | Flux-CO$_2$ | 1 | 0.484 ** | 0.726 ** | 0.758 ** |
| | Flux-CH$_4$ | | 1 | 0.497 ** | 0.485 ** |
| | air temperature | | | 1 | 0.984 ** |
| | soil temperature | | | | 1 |

** Significantly correlated at 0.01 level (bilateral); *n* = 189.

### 3.5.2. The Analysis Results of Soil Carbon Release with Soil Moisture Content

Linear, exponential, and logarithmic data fitting methods were used to fit the equation between soil carbon release and soil moisture content. The results showed a linear fitting relationship between soil methane release flux and soil moisture content, and the relationship between soil carbon dioxide release flux and soil moisture content was more in line with the exponential fitting relationship (Table 7). Although there was a linear fit between the methane release at the litter–soil interface and the soil water content after covering with different degrees of decomposition of litter, and an exponential fit relationship between the carbon dioxide release and the soil water content, there were significant differences in the fit equation between the carbon release and the soil water content under different experimental conditions.

**Table 7.** Fitting analysis of soil carbon release and soil water content.

| Project | Experiment Condition | Flux-CH$_4$ | | Flux-CO$_2$ | |
| | | Fitting Formula | R$^2$ | Fitting Formula | R$^2$ |
|---|---|---|---|---|---|
| Arbor | No litter cover | Flux = 0.0097$\omega$ + 0.0104 | 0.65 | Flux = 169.96e$^{0.050\omega}$ | 0.31 |
| | Undecomposed litter cover | Flux = 0.0073$\omega$ + 0.051 | 0.43 | Flux = 120.33e$^{0.080\omega}$ | 0.37 |
| | Semi-decomposition layer litter cover | Flux = 0.0112$\omega$ − 0.0045 | 0.43 | Flux = 411.72e$^{0.058\omega}$ | 0.54 |
| | Decomposition layer litter cover | Flux = 0.0153$\omega$ − 0.0517 | 0.42 | Flux = 257.87e$^{0.082\omega}$ | 0.51 |
| Bamboo | No litter cover | Flux = 0.0016$\omega$ + 0.0873 | 0.62 | Flux = 248.36e$^{0.034\omega}$ | 0.45 |
| | Undecomposed litter cover | Flux = 0.0038$\omega$ + 0.0519 | 0.62 | Flux = 173.02e$^{0.081\omega}$ | 0.41 |
| | Semi-decomposition layer litter cover | Flux = 0.0109$\omega$ + 0.0197 | 0.58 | Flux = 91.47e$^{0.105\omega}$ | 0.49 |
| | Decomposition layer litter cover | Flux = 0.0037$\omega$ + 0.0240 | 0.44 | Flux = 98.84e$^{0.090\omega}$ | 0.37 |

As shown in Table 8, regardless of whether the soil surface was covered with litter, there was a significant positive correlation between the methane consumption of tree forest soil and soil moisture content, with correlation coefficients of 0.548 and 0.472 ($p < 0.01$, $n = 25$), respectively. The impact of rainfall on it was insignificant, while there was a significant difference in the methane consumption of bamboo forest soil compared to it.

**Table 8.** Correlation analysis of soil carbon release with soil water content and rainfall.

| Project | | Arbor | | | | Bamboo | | | |
|---|---|---|---|---|---|---|---|---|---|
| | | Flux-CH$_4$ | Flux-CO$_2$ | Soil Moisture Content | Rainfall | Flux-CH$_4$ | Flux-CO$_2$ | Soil Moisture Content | Rainfall |
| No litter cover | Flux-CH$_4$ | 1 | 0.492 * | 0.548 ** | 0.038 | 1 | 0.201 | 0.156 | −0.198 |
| | Flux-CO$_2$ | | 1 | −0.004 | 0.066 | | 1 | 0.323 | 0.318 |
| | soil moisture content | | | 1 | 0.433 * | | | 1 | 0.178 |
| | rainfall | | | | 1 | | | | 1 |
| Covered with litter | Flux-CH$_4$ | 1 | 0.805 ** | 0.472 * | −0.009 | 1 | 0.718 ** | −0.030 | −0.222 |
| | Flux-CO$_2$ | | 1 | 0.419 * | 0.054 | | 1 | 0.454 * | −0.008 |
| | soil moisture content | | | 1 | 0.511 ** | | | 1 | 0.239 |
| | rainfall | | | | 1 | | | | 1 |

Note: ** Significant correlation at 0.01 level (bilateral); $n = 25$. * Significant correlation at 0.05 level (bilateral); $n = 25$.

## 4. Discussion

### 4.1. Comparative Analysis of Soil Carbon Release Flux under Different Types of Litter Cover

The main soil types in the research area are sandy soil, sandy loam, and sandy clay loam. Significant differences exist in soil physical and chemical properties under different types of litter cover. The organic carbon content in the surface soil of arbor and bamboo forests is 16.03 g/kg and 19.30 g/kg, respectively. Compared with arbor forests, the organic carbon content of bamboo forests is relatively high. Table 3 shows that the overall consumption of atmospheric methane by soil is higher in arbor forests than in bamboo forests, and the trend of carbon dioxide release flux is not uniform. Among them, under cover of no litter and an undecomposed layer of litter, the carbon dioxide release of bamboo forests is relatively higher than that of arbor forests. In contrast, the carbon dioxide release is lower under other conditions than that of arbor forests.

The soil in the bamboo forest is sandy loam, while the arbor forest is sandy loam and sandy clay loam. The proportion of clay particles in the soil of the arbor forest is higher, and its ability to retain water is stronger than that of the bamboo forest. The canopy layer of the arbor had a stronger ability to intercept precipitation. As a result, the soil moisture content of the arbor forest is stronger than that of the bamboo forest [35,36]. In addition, compared to arbor forests, bamboo litter belonged to the narrow and elongated type. The barrier effect of bamboo litter is relatively small. When the amount of gas blocked by the shielding effect exceeds its decomposition release, the contribution rate of litter to soil carbon release may become negative [21,37]. This phenomenon is particularly evident in bamboo forests covered with different degrees of litter decomposition, mainly because the contribution rate of litter to soil carbon release is comprehensively regulated by factors such as hydrothermal conditions, litter quality, litter input, soil animals, and microorganisms [38]. The soil carbon release under different litter treatment conditions showed significant seasonal dynamic changes [39]. Li et al. [40] have determined that removing litter would reduce soil microbial biomass by about 67% to 69%, and microbial activity would also be relatively reduced [38]. The research of Zheng et al. [41] has shown that litter removal significantly increased the soil CH$_4$ uptake in broadleaved forests by an average of 16.5%. On the contrary, double litter input decreased the soil CH$_4$ uptake by 11.1% and 20.0% in broadleaved and coniferous forests, respectively, suggesting that the function of CH$_4$ sink in forest soils decreased with additional litter input. These results originated from field observations, and the global context suggests that litter input is an important factor affecting the CH$_4$ flux uptake in forest soils, and the litter layers may reduce the atmospheric soil CH$_4$ uptake by reducing the oxygen concentration beneath the soil surface.

### 4.2. Effects of Temperature on Soil Carbon Release

Figures 4 and 6 show that, regardless of whether it was an arbor forest or bamboo forest, and whether it was covered with litter, the overall methane absorption and carbon dioxide release in the soil were highest in summer and lowest in winter. The main reason was that, during the experiment, the trend of soil temperature and temperature changes showed the highest temperature in summer and the lowest temperature in winter. The temperature difference could affect the activity of soil microorganisms, which in turn had a certain impact on the amount of soil carbon released. The rate of litter decomposition, driven primarily by microbial activity, is largely temperature dependent, suggesting that dry soil has greater litter loss than moist soil [42,43].

Grass cover in orchards is an optimal approach to enhancing the sequestration of SOC, which has been shown to improve the biological properties of soil [44]. The study of Chen et al. [28] has shown that there was a closer relationship between soil methane consumption and soil physicochemical properties, that was, soil methane consumption was significantly driven by soil environment (soil temperature, soil moisture content), soil physicochemical properties (soil organic carbon, carbon–nitrogen ratio, and $NH_4^+$ concentration). The study by Castro et al. [45] showed that, when the soil temperature was between $-5$ and $10\,^\circ$C, the consumption rate of atmospheric methane by soil was significantly positively correlated with the temperature. Dong et al. [20] found that the oxidation rate increased with the temperature increase at 0~30 $^\circ$C. Soil carbon dioxide release was positively correlated with soil temperature, moisture, and soil organic carbon [26,27] and was also regulated by soil microbial activity [28]. The correlation analysis of soil carbon dioxide release flux with temperature and soil temperature showed that, during the observation period, as the soil temperature decreased, the soil carbon dioxide release gradually decreased, and the two showed a significant positive correlation (Table 6). Decomposition processes in ecosystems have been implicated as enriching the atmosphere with $CO_2$. During leaf litter decomposition, the decomposing litter is accompanied by carbon losses. The increase in $CO_2$ resulted from the activity of soil microorganisms and fauna. $CO_2$ produced from the decomposition of organic matter is affected by temperature changes [46].

For arbor forests, the effect of temperature on carbon dioxide and methane is constantly weakened as the decomposition degree of litter deepens due to the natural barrier effect of the litter layer on soil carbon emissions. A layer of impermeable protective film was formed between the litter layer and the soil, which blocked or slowed down the gas exchange rate between the atmosphere and the soil, effectively reducing the rate of methane supply from the atmosphere to the soil [19,20]. For bamboo forests, the effects of litter with different decomposition degrees on the temperature sensitivity of soil respiration were as follows: semi-decomposed layer > undecomposed layer > decomposed layer.

### 4.3. Effects of Soil Water on Soil Carbon Release

For carbon dioxide, previous studies have shown that the changes in carbon dioxide content in dry soil were significantly influenced by soil moisture content and soil carbon dioxide flux after previous rainfall. Rainfall intensity could increase microbial consumption in moist soil, thereby altering soil carbon dioxide content [47]. Li et al. used the disaster degree correlation method to observe soil respiration in the non-growing season of the corn ecosystem for consecutive years [48]. The results showed that, when the soil moisture content was greater than 0.1 $m^3 \cdot m^{-3}$, the surface 10 cm soil moisture content showed a parabolic correlation with soil respiration, with an 18–60% correlation coefficient. Petia et al. found that temperature and soil water content significantly affected the beech forest's autotrophic respiration and heterotrophic respiration [49]. However, some studies also showed that there was not a simple linear relationship between soil water content and soil respiration [50]. Both rainfall and temperature affect the coupling relationship of soil carbon and water [51]. The correlation analysis results indicated no significant correlation between soil carbon dioxide, soil moisture content, and rainfall in tree and bamboo forests without litter cover. Still, a significant positive correlation existed between soil carbon

dioxide release and soil moisture content under litter cover. Previous studies showed that the range of increased soil respiration rate was significantly positively correlated with rainfall and with rainfall duration and average rainfall intensity. Still, the correlation was not significant [50].

From Table 7, we understood that, although there was a linear fit between the methane release at the litter–soil interface and the soil water content after covering with different degrees of decomposition of litter, and an exponential fit relationship between the carbon dioxide release and the soil water content, there were significant differences in the fit equation between the carbon release and the soil water content under different experimental conditions. Since there were significant differences in the correlation between soil water content and soil carbon release under different experimental conditions, the study's results were consistent and different from those of other studies. For example, Guo et al. found that the water–carbon fluxes in vineyards in northwest arid regions of China had a good coupling relationship, especially in the daytime. The change law of the two was obvious, and within a certain time range, there was a relatively obvious linear relationship [52]. Tu took different vegetation types from Jiufeng forest farm as the research object to conduct soil carbon-nitrogenous water coupling mechanism research. The results showed that the relationship between soil volumetric water content and soil total organic carbon, soil microbial biomass carbon, and soil respiration was relatively random, and the correlation was not significant, which should be a nonlinear coupling relationship [50]. Zhang's research showed that soil water content was significantly negatively correlated with soil organic carbon content at three sample points but not with the other three sample points [53].

The study also showed that the relationship between soil respiration and soil water content was discrete, and the correlation between the two was not significant [50,54,55]. Other relevant studies showed that soil organic carbon was positively correlated with soil water content, and soil respiration rate was significantly correlated with soil water content change [56–58]. Therefore, it was difficult to understand the relationship between soil moisture content and total organic carbon, microbial biomass carbon, and soil respiration. Soil moisture content was essential for the normal metabolism of soil microorganisms, affecting soil microbial biomass. At the same time, soil respiration, including soil microbial respiration and high or low water conditions, affects the progress of soil respiration.

### 4.4. The Synergistic Effect of Temperature and Soil Moisture

Temperature and soil moisture were key environmental factors affecting carbon flux. The results of this study indicate a significant positive correlation between soil carbon release and soil moisture content and temperature, whether in arbor or bamboo forests, and the results of this study were similar to existing research findings [22] on the coupling effect of soil moisture and temperature on soil microenvironment. Temperature directly affects the activity of soil microorganisms, while soil moisture affects soil respiration mainly by affecting soil physical and chemical properties and soil microbial activity. Within a certain range of soil water content, the $CO_2$ emission rate increased with the increase of soil water content and varied with the soil's organic carbon content. Research has also shown that 60–70% water content was most conducive to soil respiration, and soil moisture that is too low or too high will inhibit soil $CO_2$ release [59,60]. During the research period, the soil moisture content ranged from 7.3% to 35.2%. High water content would reduce soil voids and oxygen content, inhibiting soil microbial respiration and gas exchange [61]. The impact of soil moisture on soil C and nutrient dynamics could not be ignored because soil microbes mediated these processes, and fluctuations in microbial activity in response to changes in soil moisture were prevalent [62]. Tu's research showed that the relationship between soil water content and soil respiration was not a simple linear relationship, and there was a nonlinear coupling relationship between carbon and water [63].

The soil moisture content and temperature generally jointly affected the process of soil carbon release. It showed an increasing trend within a certain range with the increase of temperature and moisture, but was suppressed under extreme temperature and

moisture conditions. When the temperature was high, the soil moisture content significantly impacted the rate of soil carbon release. Similarly, when the soil moisture content was high, temperature had a greater impact on the rate of soil carbon release [64]. Schleser et al. [65] found that, when the soil moisture content was less than 75%, the increase in temperature had almost no effect on soil carbon release. However, the correlation between soil carbon release and temperature was higher when the soil moisture content increased to 100–250%. When the temperature was below 5 °C, the change in soil moisture content had almost no effect on soil carbon release. Still, when the temperature was between 10–20 °C, the correlation between soil carbon release and moisture content was higher [65].

## 5. Conclusions

(1) Regardless of whether the soil surface was covered with litter or not, soil methane flux remained negative (sink) while soil carbon dioxide flux was positive (source). The soil methane and carbon dioxide release flux showed a significant positive correlation. The impact of litter on soil carbon release flux varied significantly with the degree of litter decomposition, and the changing trend was not consistent.

(2) The fitting of the equation between soil carbon release and soil moisture content revealed a linear fitting relationship between soil methane release flux and soil moisture content, while the relationship between soil carbon dioxide release flux and soil moisture content was more in line with an exponential fitting relationship. The soil carbon release and temperature fitting results are similar to soil moisture content. However, there were significant differences in the correlation between carbon release and soil water content under different experimental conditions. Carbon dioxide, especially, was more sensitive to temperature changes.

(3) There was a mutual influence relationship between environmental factors such as soil carbon release, soil moisture content, soil temperature, and air temperature, and there were significant differences in the roles of each factor under different types of litter. The release of soil carbon was significantly positively correlated with soil moisture, soil temperature, and air temperature. In contrast, under the cover of litter, the correlation between soil carbon flux and various factors was closer.

**Author Contributions:** J.Z.: Conceptualization, Methodology, Software, Validation, Formal analysis, Investigation, Resources, Data curation, Writing—original draft preparation, Writing—review and editing, Visualization, Supervision, Project administration. T.D.: Conceptualization, Writing—review and editing. S.L.: Writing—review and editing. S.A.A.: Writing—review and editing. S.Y.: Investigation. W.L.: Resources, Writing—review and editing. T.Q.: Supervision, Writing—Review and Editing, Funding acquisition. All authors have read and agreed to the published version of the manuscript.

**Funding:** This research was supported by the Five Major Excellent Talent Programs of IWHR (WR0199A012021), the Major Science and Technology Project of the Ministry of Water Resources of the People's Republic of China (SKS-2022033) and the National Science Fund Project (Grant No. 52130907).

**Data Availability Statement:** The original contributions presented in the study are included in the article, further inquiries can be directed to the corresponding author.

**Conflicts of Interest:** The authors declare no conflicts of interest.

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
