# Peer review of "Carbon Release Characteristics at Soil–Air Interface under Litter Cover with Different Decomposition Degrees in the Arbor and Bamboo Forests of Pi River Basin"

_land, doi:10.3390/land13040427_

Round 1
Reviewer 1 Report
Comments and Suggestions for Authors
The article is aimed at studying the terrestrial carbon cycle in the arbor and bamboo forests of Pi River Basin. The authors identified the characteristics of carbon release at litter-soil interface and the rule of carbon release at soil-air interface during litter decomposition. The research is relevant and of theoretical relevance for predicting the response of terrestrial carbon cycle to future climate change. The article is qualitatively written and well illustrated.
Title
The title corresponds to the content.
Introduction
My advice would be to emphasize the novelty of the research and justify your choice of study area.
Materials and Methods
Methodological approaches are not fully described. It is better to give a more detailed description of the plant communities and soils being studied and it is better to present this in the form of a table. The authors did not indicate what type of soil was under what forests and the taxation characteristics of the forest types. Authors should add a subsection on statistical data processing. In my opinion, it is worth additionally bringing the soil taxonomy in accordance with the soil classification systems (FAO, WRB).
Results
The results of the study are presented quite clearly and illustrated with 4 figures and 6 tables.
Section 3.3. It is necessary to indicate for what period the data of the studied parameters are averaged: during the study (15 months) or over a year.
Discussion
The authors have discussed results quite well. However, it would be good to add more modern research (no more than five years old) into the discussion.
Conclusions
Conclusions follow from the results and are reasonable.
There are minor errors in the formatting of the article:
- The list of references has double numbering.
- Remove dots at the end of sentences in the names of figures and tables.
Author Response
Dear reviewer:
Thank you very much for taking the time to review this manuscript. Please find the detailed responses below and the corresponding revisions/corrections highlighted/in track changes in the re-submitted files.
Please see the attachment.

Reviewer 2 Report
Comments and Suggestions for Authors
The study examines an important research topic related to carbon cycling and the effects of litter decomposition. The methods and analyses seem technically sound.
- The writing quality is good overall but there are some minor grammatical/phrasing issues in parts that should be addressed.
- More context about the relevance of the research and how it builds on previous studies is needed, especially in the Introduction and Discussion sections.
Abstract:
- Consider revising the last sentence to more specifically indicate the relevance of the findings.
The sentence “The results showed that regardless of whether the soil surface was covered with litter or not...” is unclear. Be more specific, such as “Results showed that soil methane flux remained negative (sink) while soil CO2 flux was positive (source) in both litter-covered and bare soil conditions”.
Introduction:
- Should cite previous studies on this specific topic to identify key knowledge gaps the current study aims to address.
- A more clearly stated hypothesis or research objective is needed.
The sentence starting “It was the main source...” seems out of place here. Consider moving this to the next paragraph.
This sentence “The quantity and quality of litter could alter soil microbial respiration rate...” comes out of nowhere. Provide some context first before stating this or cite a reference.
Methods:
- Were soil parameters like texture, organic matter content measured? This could help interpret differences between forest types.
Specify details such as make/model for the greenhouse gas analyzer used. Also state statistical analysis methods here.
- Statistical analysis methods should be described.
Results:
- Add some test statistics (p values, error bars etc.) to indicate significance of results.
Add error bars or some other indicator of variability to Figure 4 & 6. Also include test statistics and p-values for the fluxes under different litter conditions.
Discussion:
- Should more directly address how the findings align or contrast with cited studies and discuss possible reasons.
- Limitations of the experimental design and analyses should be considered.
When comparing soil carbon fluxes to other regions, cite references to provide context for the values you are comparing against.
The statement “Studies had shown that there was a closer relationship between soil methane consumption and soil physicochemical properties...” is vague. Be more specific about which studies and parameters.
Please consider cite this paper, Grass cover increases soil microbial abundance and diversity and extracellular enzyme activities in orchards: A synthesis across China.
Conclusions:
- Should focus more on the broader significance and implications of the research.
Author Response

(The authors gave the same response as above.)

Reviewer 3 Report
Comments and Suggestions for Authors
Review for “Carbon release characteristics at soil-air interface under litter cover with different decomposition degrees in the arbor and bamboo forests of Pi River Basin”
Overall, the presented work contains the results of a year-long experiment to assess the influence of litter on the fluxes of methane and carbon dioxide from the soil surface in various forest types. The work was performed using standard methods. After significant revision it can be published.
General remarks
1. There are repetitions in the text; I have pointed out some of them; they need to be checked carefully.
2. The paper presents calculations of correlations of C fluxes with soil moisture and temperature. But the data themselves are missing, which makes perception extremely difficult. It is necessary to provide this data, preferably in the form of graphs. And, be sure to give the methods by which they were obtained.
3. There are also frequent references to weather conditions (rain events, for example). These data for the year of measurement could also be presented.
4. The Discussion section needs to be revised. In this form it looks more like an introduction. It is necessary not only to present literature data that are generally well known (for example, the effect of moisture and temperature on emissions), but to compare your data and conclusions with those of other researchers.
5. In the conclusion, you should clearly write what new was obtained in the study and how it agrees with the known facts.
6. English language needs to be improved.
Remarks
Abstract
L 18 Replace the word " box" with the word "chamber"
1. Introduction
It is not clear why the introduction is written in the past tense. Needs to be adjusted (remade at present tense).
L 40 “litter-soil” will be better
Figure 1. Rectangle with layers of litter is not readable
|
2. Materials and methods |
L 116-117 I propose to reformulate the phrase, for example using the following expression: “ventilate the chamber by turning the open part up”.
Replace the words “sampling box”, “flux box” with the word "chamber"
It is advisable to indicate the frequency of measurements
3. Results
3.1.
L 154-155 somewhere there's an extra word "without"
Figure 5 The drawing requires clarification. The designations of the vertical axes in Figures 4 and 5 are the same, although the essence is different. In Figure 5, this is not a flow, but a delta between the flow in the area without litter and with litter. In accordance with this, we also recommend clarifying the caption of Figure 5. Confusion is also caused by the different color and symbol designation of different litters in Figures 4 and 5. It is necessary to unify.
3.2.
L 239-240 Repeating a phrase from the introduction (L 68-69)
Figure 7 Same wishes as for Figure 5
3.3.
L 253 How does the data on average methane consumption compare with the data on the L 154? Is it the same? The same L 255. All data from Table 1 is presented in Section 3.1.You need to change the text to avoid repetitions.
Table 2 needs to be rearranged into section 4, where it is mentioned.
4. Discussion
314 “The soil in the bamboo forest was sandy soil, while the upper layer of the soil in the arbor forest was sandy soil” Are both soils sandy?
L 417 It is better to use the term “moisture” to characterize the soil.
Section 4.4. reads more like an introduction than a discussion. What was the soil moisture range in your study? In this form (without comparison with your own data), this part should be transferred to the introduction.
5. Conclusion
The conclusion needs to be redone. It should not look like a summary of the text of the article. The main theme of your paper is the influence of litter on carbon fluxes. There is no mention of this in the conclusion at all. Why do you describe the effect of moisture, but don’t write about temperature? And so on. In conclusion, try to give the main findings based on the tasks assigned and include something new that you have managed to get.
Comments on the Quality of English LanguageEnglish language needs to be improved
Author Response

(The authors gave the same response as above.)

Reviewer 4 Report
Comments and Suggestions for Authors
The manuscript uses a space-for-time approach to quantifiy the methane and carbon dioxide release from soil covered with differently decomposed litter.
Even though it has an intrinsic value in itself to quantify carbon fluxes from litter, I miss the larger framework of this study. Why is it important to look at the litter layer in different vegetation types and what conclusions and management implications for the future can be drawn from the results?
This lack can already be seen in the abstract. An introduction is missing here completely and also a discussion or conclusion in not there, only results are presented in the abstract.
The study design is not shown properly. The different litter variants are not mentioned in the methods section at all, even though they represent the treatments. There is also no information on the number of stands that were investigated. It seems that you are dealing with an n = 1 for arbor or bamboo stands. There is also no information in the methods section on statistical analysis.
In addition, information on the investigated species is missing. What kind of tree species dominates in the forest that was studied? What bamboo species is present?
Due to the missing information in the methods I do not understand the difference between Figure 4 and 5 or between Figure 6 and 7. Do I understand it right, that extra litter was added?
Mentioning of figures and tables in the text is often missing.
In the end, I am missing a real conclusion what results might imply for management or for aspects of carbon sequestration and climate change.
Comments on the Quality of English LanguageLanguage is ok, however I do not understand the use of past tense in the introduction. It should read: Litter is an important component…
Author Response

(The authors gave the same response as above.)

Round 2
Reviewer 3 Report
Comments and Suggestions for Authors
The authors did a good job with the manuscript. I have no major comments